# Placental Cyclophilin A Expression in Pregnancies Complicated with Hypertension

**DOI:** 10.3390/ijerph19095448

**Published:** 2022-04-29

**Authors:** Elia Shazniza Shaaya, Azyani Yahaya, Muaatamarulain Mustangin, Nurwardah Alfian, Azimatun Noor Aizuddin, Yin Ping Wong, Geok Chin Tan

**Affiliations:** 1Department of Pathology, Faculty of Medicine, Universiti Kebangsaan Malaysia, Kuala Lumpur 56000, Malaysia; elia_shaz@yahoo.com (E.S.S.); azyani@ppukm.ukm.edu.my (A.Y.); amar@ppukm.ukm.edu.my (M.M.); nurwardah@ppukm.ukm.edu.my (N.A.); 2Department of Community Health, Faculty of Medicine, Universiti Kebangsaan Malaysia, Kuala Lumpur 56000, Malaysia; azimatunnoor@ppukm.ukm.edu.my

**Keywords:** cyclophilin A, placenta, hypertension, perinatal outcomes

## Abstract

Introduction: Cyclophilin A was reported to be increased in the serum of mothers with preeclampsia, and is implicated in its pathogenesis. This study aimed to determine the expression of cyclophilin A in the placenta of mothers with and without hypertension, and to correlate its expression with maternal complications and adverse perinatal outcomes. Materials and Methods: This study consisted of a total of 70 cases (35 cases of mothers with hypertension, and 35 normotensive mothers as a control). Cyclophilin A immunohistochemistry was performed on a paraffin-embedded tissue section of placenta submitted at full thickness in order to evaluate the expression in fetal endothelial cells, cytotrophoblasts, syncytiotrophoblasts, maternal endothelial cells and decidual cells. The cyclophilin A expression was scored as weak, moderate or strong intensity. Results: The hypertensive group was more likely to have preterm deliveries (*p* < 0.0001), caesarean sections (*p* < 0.0001), and infants admitted to the intensive care unit (*p* < 0.001). Fifty-one percent of the fetal endothelial cells and cytotrophoblasts expressed cyclophilin A in the hypertensive group, compared to only 28.6% in the normotensive group. However, the difference was not statistically significant (*p* = 0.086). Conclusion: We found no significant difference in placental cyclophilin A expression between hypertensive and normotensive mothers. There was also no difference in expression in mothers with and without maternal complications and adverse perinatal outcomes.

## 1. Introduction

Hypertensive disorders in pregnancy (HDP) affect approximately 2–10% of all pregnancies [1,2], and are one of the leading causes of maternal and perinatal morbidity and mortality, including preeclampsia, eclampsia, HELLP syndrome, placental abruption, fetal growth restriction, prematurity and stillbirth [3,4,5]. Our previous studies showed the increased expression of endothelial cell-specific molecule 1 and vascular endothelial growth factor; both related vascular developments were increased in the placenta of mothers with hypertension [6,7]. Preeclampsia is a disease of vascular maldevelopment. It develops in two stages. In the first stage, there is abnormal placentation with ineffective cytotrophoblast invasion of the uterine spiral arteries, leading to inadequate spiral artery remodelling, the narrowing of the maternal blood vessels, and reduced blood supply to the placenta. In the second stage, the placental ischaemia induces a cascade of inflammatory events and angiogenic imbalance that cause endothelial dysfunction [4,5,8].

Cyclophilin A (CyPA) is a ubiquitous and highly conserved protein, exhibiting peptidyl prolyl isomerase activity [9]. It regulates many biological processes such as intracellular signalling, inflammation and apoptosis. Studies have demonstrated that CyPA is involved in many pathophysiological mechanisms of cardiovascular diseases, such as hypertension, atherosclerosis, atherosclerotic aortic aneurysm, peripheral arterial occlusion, pulmonary arterial hypertension and myocardial hypertrophy [9,10,11,12], as well as inflammatory conditions such as rheumatoid arthritis [12,13]. Satoh et al. (2008) reported that CyPA mediates vascular remodelling by promoting inflammation and vascular smooth muscle proliferation [12]. Meanwhile, Hu et al. (2020) found that CyPA inhibited the migration and invasion of trophoblasts, and suggested that it may be involved in the development of preeclampsia [8].

CyPA is found in abundance intra- and extra-cellularly [9]. Intracellular CyPA has catalytic and chaperone activities of PPIase, and is involved in protein folding, protein trafficking and protein activation [9,12]. In response to oxidative stress, CyPA is secreted extracellularly by many types of cells, such as vascular smooth muscle cells, endothelial cells, and macrophages. The extracellular CyPA (eCyPA) activates proinflammatory pathways via the production of cytokines. In addition to the functions of intracellular CyPA, such as inflammation and proliferation, eCyPA also possesses other functions, such as apoptosis, cellular proliferation and migration, matrix degradation by the activation of matrix metalloproteinases, and the generation of reactive oxygen species [9,12].

Chang et al. (2013) reported that the serum levels of CyPA were significantly higher in individuals with hypertension compared to healthy subjects, and suggested that CyPA may be used a biomarker for the assessment of hypertension [10]. There are multiple studies that have described an increased serum level of CyPA in mothers with gestational hypertension and preeclampsia [13,14,15,16]. Wang et al. (2018) [13] and Celik et al. (2020) [14] found that increased serum CyPA in first- and early second-trimester pregnancy increases the risk of the subsequent development of preeclampsia. Notably, the expression of CyPA in the placenta of mothers with hypertension is still largely not known. So far, only two studies have reported that placental CyPA expression is higher in pre-eclampsia than in normal placenta [8,16]. They found that CyPA was expressed in cytotrophoblasts, syncytiotrophoblasts and vascular endothelial cells. Further evaluation is needed in order to validate this finding.

Recent studies have described the potential of drugs specific to Cyclophilin A that may be used in cardiovascular diseases. Hence, this study could reveal more information on the possible usage of Cyclophilin A as a biomarker for the prediction of disease and the development of a new therapeutic target. The aim of this study is to determine the expression of CyPA in the placenta of mothers with and without hypertension, and to correlate its expression with maternal and perinatal outcomes.

## 2. Materials and Methods

This was a retrospective on archival histopathological materials over a period of 3 years. Clinical information was retrieved from a medical record unit. The study consisted of 70 cases (35 placenta of mothers with hypertension and 35 placenta of normotensive mothers). All of the placenta were reviewed, and one full-thickness section was selected for CyPA immunohistochemical study. Hypertension in pregnancy is defined as BP ≥140/90 mmHg after 20 weeks of gestation), and the control group consisted of normotensive pregnant women beyond 20 weeks of gestation. The pregnancies of mothers with chronic hypertension, diabetes mellitus, renal disorders and ischaemic heart disease were excluded from the study. Cases with incomplete clinical history were also excluded. This study was approved by our institutional ethics committee, Universiti Kebangsaan Malaysia, Cheras Campus, Kuala Lumpur, Malaysia (Ref: UKM PPI/111/8/JEP-2020-191).

### 2.1. CyPA Immunohistochemistry

Anti-Cyclophilin A antibody [EPR7511] (Code Number: ab126738, Abcam, Cambridge, UK) was used at a dilution of 1:300. Human breast carcinoma tissue was used as the positive control tissue. Cytoplasmic staining was considered as positive. Immunohistochemical staining was performed on the tissue sections using the protocol from the Envision FLEX Mini Kit, at a high pH (Link) (Code Number: K8023, Dako Agilent Denmark, Glostrup, Denmark). Primary antibody was diluted to the optimal dilution using Antibody Diluent, Dako REALTM (Code Number: S0809, Dako Agilent Denmark). The washing steps between each reagent were performed using Envision FLEX Wash Buffer 20× (Code Number: K8007, Dako Agilent Denmark) diluted to a working solution with deionized water. The DAB-containing Substrate Working Solution was prepared by diluting the concentrated Envision FLEX DAB+ Chromogen with Envision FLEX Substrate Buffer (Code Number: K8023, Dako Agilent Denmark).

Tissue blocks were sectioned at 3 µm thickness and mounted on an adhesive glass slide, Platinum Pro White (Product Number: PRO-01, Matsunami, Kishiwada, Japan). The slides were left to be air-dried at room temperature overnight. The tissue slides were then incubated on a hot plate at 60 °C for 1 h. An initial deparaffinization and pre-treatment step was performed in a Decloaking Chamber™ NxGen (Reference Number: DC2012-220V, Biocare Medical California, Pacheco, CA, USA) using the Envision FLEX Target Retrieval Solution, at a high pH (Code Number: DM828, Dako Agilent Denmark), with the conditions of a temperature of 110 °C and a time of 30 min, followed by cooling at room temperature for 30 min and rinsing with running tap water for 3 min. The slides were subsequently incubated with Envision FLEX Peroxidase-Blocking Reagent (Code Number: DM821, Dako Agilent Denmark) for 10 min, followed by a washing step.

The slides were then incubated with primary antibody for 30 min at room temperature, followed by washing step. Subsequently, the slides were incubated with Envision FLEX HRP (Code Number: DM822, Dako Agilent Denmark) for 30 min, followed by washing step. After that, the slides were incubated with DAB-containing Substrate Working Solution for 7 min. They were then counterstained with Hematoxylin 2 (Reference Number: 7231, Thermo Scientific, Waltham, MA, USA) for 5 s after the procedures have been completed followed by dehydration step with increasing alcohol solutions (80%, 90%, 100% and 100%) and 2-times Xylene. Finally, the slides were mounted using CoverSeal™-X Mounting Medium (Catalogue Number: FX2176, Cancer Diagnostics, Durham, NC, USA).

### 2.2. CyPA Immunostaining Analysis

The expression of Cyclophilin A was evaluated in fetal endothelial cells, cytotrophoblasts, syncytiotrophoblasts, maternal endothelial cells and decidual cells. The staining intensity was scored on a scale of 0–3 (0—no staining, 1—weak (1+), 2—moderate (2+) and 3—strong (3+)). Strong expression (3+) with a percentage of positive cells of ≥75% was considered as positive for the statistical analysis. All of the cases were evaluated by two independent pathologists. A consensus was found in cases where there was discrepancy.

### 2.3. Statistical Analysis

The data analysis was performed using SPSS version 26.0 (New York, NY, USA) and a GraphPad online calculator (https://www.graphpad.com/quickcalcs/contingency2/ (accessed on 1 February 2022). The differences in Cyclophilin A expression in the placenta between hypertensive and normotensive subjects were evaluated using a Pearson chi-square test. A *p* value of <0.05 was considered to be statistically significant.

## 3. Results

### 3.1. Maternal Clinical Characteristics

This study consisted of a total of 70 cases comprises of 35 cases of mothers with hypertension in pregnancy and 35 cases of normotensive mothers. The gestational age at delivery of the hypertensive group and normotensive group ranged between 26 to 40 weeks (an average of 35.1 weeks) and 37 to 41 weeks (an average of 38.8 weeks), respectively. The mothers with hypertension had an earlier gestation at delivery compared to the normotensive mothers, and the difference was statistically significant (*p* < 0.0001). Nine of the mothers (9/35, 25.7%) with hypertension were >35 years old. In contrast, six of the mothers (6/35, 17.1%) with normal blood pressure were >35 years old (*p* = 0.56). The majority of the mothers with hypertension (30/35, 85.7%) underwent lower-segment caesarean section, while only 17.1% (6/35) of the normotensive mothers underwent lower-segment caesarean section (*p* < 0.0001) (Table 1). 

### 3.2. Adverse Perinatal Outcomes between Hypertensive and Normotensive Mothers

We found that 57.1% (20/35) of the mothers with hypertension had premature infants, compared to only 2.9% (1/35) of the normotensive mothers having premature infants (*p* < 0.001). In addition, the neonate’s APGAR score at 1 min and 5 min in the hypertensive group were <7, were 25.7% (9/35) and 11.4% (4/35). On the contrary, none of the neonates in the normotensive group had an APGAR score <7. The *p* value for the APGAR scores at 1 min and 5 min were 0.001 and 0.039, respectively. About half (48.6%, 17/35) of the neonates in the hypertensive group were admitted to a neonatal intensive care unit, while none were admitted in the normotensive group (*p* < 0.001).

### 3.3. Cyclophilin A Expression in Different Types of Placental Cells

CyPA was expressed in both the nucleus and cytoplasm of the fetal endothelial cells, cytotrophoblasts, maternal endothelial cells and decidual cells in the placenta. Notably, it was not expressed in syncytiotrophoblasts (Figure 1). In the placenta of mothers with hypertension, the strong expression of CyPA was observed in 51.4% (18/35) of both the fetal endothelial cells and cytotrophoblasts. In contrast, only 28.6% (10/35) of the normotensive group showed strong CyPA expression. However, the difference was not statistically significant (*p* = 0.0869) (Table 2). Meanwhile, in the maternal endothelial cells, the expression of CyPA in hypertensive and normotensive groups was seen in 62.9% (22/35) and 57.1% (20/35) of cases, respectively (*p* = 0.235). Strong expression was observed in the decidual cells of all cases, in both the hypertensive (35/35, 100%) and normotensive (35/35, 100%) groups. CyPA expression was completely absent in the syncytiotrophoblasts of all cases in both the hypertensive (0/35) and normotensive (0/35) groups (Table 2 and Appendix A).

### 3.4. Cyclophilin A Expression with Maternal Complications and Adverse Perinatal Outcomes in Hypertensive Disorders of Pregnancy

In this study, two of the mothers with hypertension developed eclampsia, two had HELPP syndrome, one had acute renal injury, and one succumbed to maternal mortality. We compared the expression of cyclophilin A in these patients. There was no statistically significant difference in CyPA expression between the two groups for acute renal injury, eclampsia, HELLP syndrome or maternal death (Table 3). The expression of CyPA was compared between cases with and without adverse perinatal outcomes such as stillbirth, preterm delivery, an APGAR score <7 at 1 min and 5 min, and admission to a neonatal intensive care unit (NICU). There was no significant difference in CyPA expression in the various placental cell types in adverse perinatal outcomes (Table 3). Although the CyPA expression in the placenta of neonates with an APGAR score of <7 at 1 min was 100% (9/9) compared to only 69.2% (18/26) in APGAR >7, it was not quite statistically significant (*p* = 0.0815).

### 3.5. Cyclophilin A and Placental Histological Changes Related to Hypertension

The histological features associated with hypertension in pregnancy include acute atherosis and the retention of the smooth muscle of the maternal blood vessels, and increased syncytial knot formation, also known as Tenney Parker change (Figure 2). Our study showed no significant difference in the CyPA expression in all the three histological features. CyPA was strongly expressed in 80% (8/10), 80% (8/10) and 88.9% (8/9) of fetal endothelial cells, cytotrophoblasts and maternal endothelial cells, respectively, in the placenta of mothers with acute atherosis, compared to 76% (19/25), 76% (19/25) and 86.9% (20/23) in the placenta of mothers without acute atherosis (*p* = 1.0). Similarly, there were no significant differences in CyPA expression in the retention of smooth muscle in the maternal vessel walls or increased syncytial knot formation (Table 4).

## 4. Discussion

In this study, we found that the gestational age at delivery was significantly earlier in the hypertensive group compared to the normotensive group. Similarly, previous studies reported that hypertension in pregnancy was associated with preterm labour [4,14,15,16]. Lower-segment caesarean section was the most frequent mode of delivery in the hypertensive group compared to the normotensive group. In addition, in the hypertensive group, the infants were more frequently premature, had a lower APGAR score at both 1 and 5 min, and were more likely to be admitted to the neonatal intensive care unit. Celik et al. (2020) and Sun et al. (2019) reported similar result to our study, in which the APGAR score at 1 min was significantly lower in the preeclampsia group compared to the normotensive group [14,16].

Studies have shown that the serum CyPA level is increased in gestational hypertension and preeclampsia, suggesting that CyPA may play a role in the pathogenesis of hypertension in pregnancy [13,14,15,16]. Currently, there are limited data on CyPA expression in the placenta. There were only two studies describing the presence of CyPA expression in the placenta of mothers with hypertension [8,17]. Hu et al. (2020) also showed that CyPA inhibited the migration and invasion of trophoblasts, and suggested its involvement in the development of preeclampsia [8]. Sun et al. (2019) reported that the CyPA mRNA level in placenta was significantly increased in subjects with preeclampsia compared to healthy pregnancies. They showed that CyPA was expressed in the cytoplasm of villous trophoblasts and the extracellular matrix with higher expression observed in the placenta of severe preeclampsia patients [16].

Hu et al. (2020) also used the immunohistochemistry method to evaluate the expression of CyPA in placenta. Similar to our study, they showed that CyPA is widely distributed within the placenta [8]. Our study showed that CyPA was consistently expressed in all of the decidual cells of both mothers with and without hypertension. Meier et al. (1995) described the detection of CyPA protein in human decidua in the first trimester of pregnancy [17]. The fetal endothelial cells and cytotrophoblasts showed higher CyPA expression in the hypertensive group. However, it was not statistically significant. On the contrary, Hu’s team described a higher CyPA expression in the placenta of mother with severe preeclampsia, of which the staining was in the cytotrophoblasts, syncytiotrophoblasts and vascular endothelial cells. Curiously, we found that syncytiotrophoblasts do not express Cyclophilin A in the placenta of mothers with and without hypertension (see Table 2), which is again in contrast with the study by Hu et al. (2020) [8].

Wibowo et al. (2017) studied the serum CyPA levels in both the maternal and cord blood of severe preeclampsia and normal pregnancies. Interestingly, they found that subjects with severe preeclampsia had higher average CyPA levels in the maternal serum compared with normotensive pregnancy. However, the average CyPA level in the cord blood in subjects with severe preeclampsia was lower than that in normotensive pregnancies [15]. This could explain why we did not find an increase in the CyPA expression in the placenta of subjects with hypertension in pregnancy. The CyPA may be produced and elevated mainly in the mother. However, further study is required in order to ascertain this.

The histological features of hypertension in placenta include acute atherosis with presence of the fibrinoid necrosis of the wall of the maternal vessels, an increase in the number of the syncytial knot formations (Tenney Parker change), and the retention of smooth muscle in the wall of the maternal vessels [6,18,19,20]. Kim et al. (2015) reported that the identification of acute atherosis was associated with a more severe disease and small for gestation babies [21]. There was no significant difference in CyPA expression in the placenta of the hypertensive group with and without the histological features of hypertension. Surprisingly, we found that CyPA expression was not increased in the hypertensive group with maternal complications or adverse perinatal outcomes. However, this could be due to the limited number of cases with maternal complications and stillbirth.

Our study may be limited by the small number of cases. Even though the expressions of CyPA were not significantly different between the hypertensive and normotensive groups, the *p* values for fetal endothelial cells and cytotrophoblasts were close to 0.05 (*p* = 0.0869). A large study is warranted in order to further confirm our findings.

## 5. Conclusions

CyPA was expressed in the nucleus and cytoplasm of the fetal endothelial cells, cytotrophoblasts, maternal endothelial cells and decidual cells of the placenta, but not in the syncytiotrophoblasts. There was no significant difference in CyPA expression between the hypertensive and normotensive group, although a slightly higher expression was observed in fetal endothelial cells and cytotrophoblasts. There was no statistically significant difference in CyPA expression in the hypertensive group with and without maternal complications or adverse perinatal outcomes. Our findings suggest that CyPA may have a more significant role in the maternal circulation than in the placenta.

## Figures and Tables

**Figure 1 ijerph-19-05448-f001:**
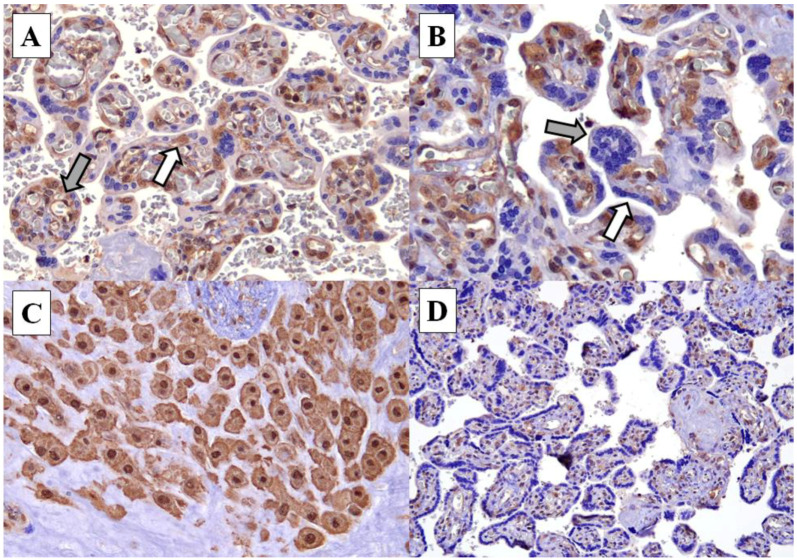
(**A**) Cyclophilin A expression in fetal endothelial cells (grey arrow) and cytotrophoblasts (white arrow) of the chorionic villi (×200). (**B**) Absence of cyclophilin A expression in the syncytiotrophoblasts (white arrow) and syncytial knots (grey arrow) (×400). (**C**) The decidual cells demonstrate strong Cyclophilin A expression (×400). (**D**) Weak cyclophilin A staining in fetal endothelial cells and cytotrophoblasts (×100).

**Figure 2 ijerph-19-05448-f002:**
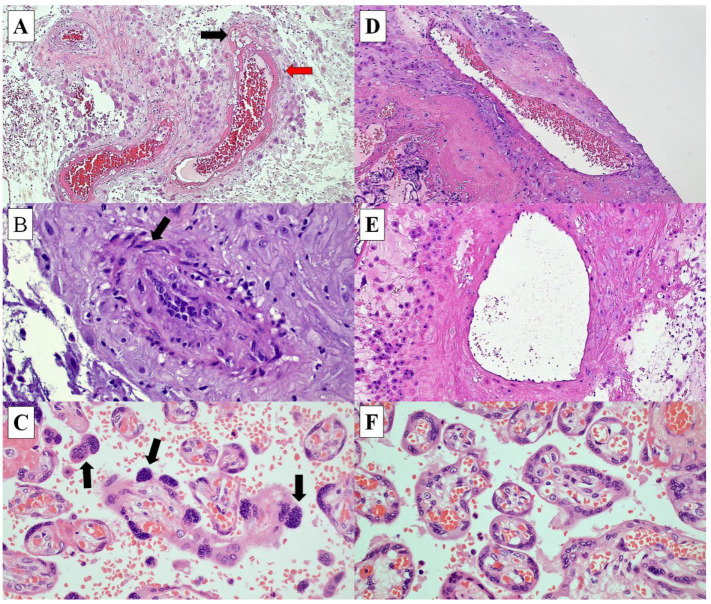
The histological features of placenta from pregnancy with hypertension (**A**–**C**) and without hypertension (**D**–**F**). (**A**) Maternal blood vessels demonstrating fibrinoid necrosis (red arrow) and foam cells (black arrow) (acute atherosis) (H&E, ×200). (**B**) Maternal blood vessels with the retention of smooth muscle/mural hypertrophy (arrow) (H&E, ×400). (**C**) Chorionic villi with increased syncytial knot formations (arrows) (H&E, ×400). (**D**,**E**) Normal histological appearance of maternal blood vessels (H&E, ×200 (**D**), ×400 (**E**)), and (**F**) chorionic villi (H&E, ×400).

**Table 1 ijerph-19-05448-t001:** Comparison of the maternal clinical parameters and perinatal outcomes between mothers with and without hypertension in pregnancy.

Maternal Clinical Parameters	Hypertensive *n* (%)	Normotensive *n* (%)	*p* Value
Age			
<35 years old	26 (74.3)	29 (82.9)	0.56
>35 years old	9 (25.7)	6 (17.1)	
Gestational age ^⸸^			
Range (weeks)	26–40	37–41	
Mean (weeks)	35.1	38.8	<0.0001 *
Median (weeks)	36	39	
SD	4.0	1.2	
Hypertension in previous pregnancy			
Yes	4 (11.4)	0 (0)	0.11
No	31 (88.6)	35 (100)	
Type of delivery			
SVD	5 (14.3)	28 (80.0)	<0.0001 *
Lower segment caesarean section	30 (85.7)	6 (17.1)	
Instrumental (forceps/vacuum)	0 (0.0)	1 (2.9)	
**Adverse perinatal outcomes**			
Stillbirth			
Yes	34 (97.1)	35 (100)	0.314
No	1 (2.9)	0 (0)	
Prematurity			
Yes	20 (57.1)	1 (2.9)	<0.001 *
No	15 (42.9)	34 (97.1)	
APGAR score at 1 min			
≥7	26 (74.3)	35 (100)	0.001 *
<7	9 (25.7)	0 (0)	
APGAR score at 5 min			
≥7	31(88.6)	35 (100)	0.039 *
<7	4 (11.4)	0 (0)	
Admission to nicu			
Yes	17 (48.6)	0 (0)	<0.001 *
No	18 (51.4)	35 (100)	

SD, Standard deviation; SVD, spontaneous vaginal delivery; nicu, neonatal intensive care unit. * A *p* value of <0.05 was considered as significant; ^⸸^ the gestational age is in weeks instead of the number of cases (*n*).

**Table 2 ijerph-19-05448-t002:** Comparison of the cyclophilin A expression between hypertensive and normotensive groups in maternal endothelial cells, fetal endothelial cells, syncytiotrophoblasts, cytotrophoblasts, and decidual cells.

Placental Cell Types	Cyclophilin A Expression	Hypertensive (*n* = 35) (%)	Normotensive (*n* = 35) (%)	*p* Value
Fetal endothelial cells	Positive	18 (51.4)	10 (28.6)	0.0869
	Negative	17 (48.6)	25 (71.4)	
Cytotrophoblast	Positive	18 (51.4)	10 (28.6)	0.0869
	Negative	17 (48.6)	25 (71.4)	
Maternal endothelial cells	Positive	22 (62.9)	20 (57.1)	0.235
	Negative	10 (28.4)	7 (20.0)	
	NA	3 (8.6)	8 (22.9)	
Syncytiotrophoblast	Positive	0	0	1.0
	Negative	35	35	
Decidual cells	Positive	35	35	1.0
	Negative	0	0	

NA—not available.

**Table 3 ijerph-19-05448-t003:** Comparison of the cyclophilin A expression in the placenta of hypertensive mothers with maternal complications and adverse perinatal outcomes.

	Cyclophilin A Expression									
Maternal Complications	Fetal Endothelial Cells		*p* Value	Cytotrophoblasts		*p* Value	Maternal Endothelial Cells			*p* Value
	Positive *n* (%)	Negative *n* (%)		Positive *n* (%)	Negative *n* (%)		Positive *n* (%)	Negative *n* (%)	NA	
Eclampsia										
Yes	1 (50)	1 (50)	0.4101	1 (50)	1 (50)	0.4101	2 (100)	0 (0)	0 (0)	1.0
No	26 (78.8)	7 (21.2)		26 (78.8)	7 (21,2)		26 (78.8)	4 (12.2)	3 (10.0)	
Acute kidney injury										
Yes	0 (0.0)	1 (100)	0.2286	0 (0.0)	1 (100)	0.2286	1 (100)	0 (0)	0 (0)	1.0
No	27 (79.4)	7 (20.6)		27 (79.4)	7 (20.6)		27 (79.4)	4 (11.8)	3 (8.8)	
HELPP										
Yes	2 (100)	0 (0.0)	1.0	2 (100)	0 (0.0)	1.0	1 (50.0)	0 (0)	1 (50.0)	1.0
No	25 (75.8)	8 (24.2)		25 (75.8)	8 (24.2)		27 (81.8)	4 (12.1)	2 (6.1)	
Maternal mortality										
Yes	1 (100)	0 (0.0)	1.0	1 (100)	0 (0.0)	1.0	1 (100)	0 (0)	0 (0)	1.0
No	26 (76.5)	8 (23.5)		26 (76.5)	8 (23.5)		27 (79.4)	4 (11.8)	3 (8.8)	
**Adverse perinatal outcomes**										
Stillbirth										
Yes	1 (3.7)	0 (0.0)	1.0	1 (3.7)	0 (0.0)	1.0	1 (3.6)	0 (0.0)	0 (0.0)	1.0
No	26 (76.5)	8 (23.5)		26 (76.5)	8 (23.5)		27 (79.4)	4 (11.8)	3 (8.8)	
Prematurity										
Yes	15 (75.0)	5 (25.0)	1.0	15 (75.0)	5 (25.0)	1.0	16 (80.0)	3 (15.0)	1 (5.0)	0.545
No	12 (80.0)	3 (20.0)		12 (80.0)	3 (20.0)		12 (80.0)	1 (6.7)	2 (13.3)	
APGAR score at 1 min										
≥7	18 (69.2)	8 (30.8)	0.0815	18 (69.2)	8 (30.8)	0.0815	22 (84.6)	3 (11.5)	1 (3.9)	1.0
<7	9 (100)	0 (0)		9 (100)	0 (0)		6 (66.7)	1 (11.1)	2 (22.2)	
APGAR score at 5 min										
≥7	23 (74.2)	8 (34.8)	0.5531	23 (74.2)	8 (34.8)	0.5531	26 (83.9)	3 (9.7)	2 (6.5)	0.3395
<7	4 (100)	0 (0)		4 (100)	0 (0)		2 (50.0)	1 (25.0)	1 (25.0)	
Admission to nicu										
Yes	12 (70.6)	5 (29.4)	0.4430	12 (70.6)	5 (29.4)	0.4430	14 (82.4)	2 (11.8)	1 (5.8)	1.0
No	15 (83.3)	3 (16.7)		15 (83.3)	3 (16.7)		14 (77.8)	2 (11.1)	2 (11.1)	

nicu—neonatal intensive care unit.

**Table 4 ijerph-19-05448-t004:** Comparison of the cyclophilin A expression and placental histological changes associated with hypertension.

	Cyclophilin A Expression									
	Fetal Endothelial Cells		*p* Value	Cytotrophoblasts		*p* Value	Maternal Endothelial Cells			*p* Value
	Positive *n* (%)	Negative *n* (%)		Positive *n* (%)	Negative *n* (%)		Positive *n* (%)	Negative *n* (%)	NA	
Acute atherosis										
Yes	8 (80.0)	2 (20.0)	1.0	8 (80.0)	2 (20.0)	1.0	8 (80.0)	1 (10.0)	1 (10.0)	1.0
No	19 (76.0)	6 (24.0)		19 (76.0)	6 (24.0)		20 (71.4)	3 (75.0)	2 (66.7)	
Retention of smooth muscles in the maternal wall										
Yes	6 (85.7)	1 (14.2)	1.0	6 (85.7)	1 (14.2)	1.0	4 (57.1)	1 (14.3)	2 (28.6)	0.512
No	21 (75.0)	7 (25.0)		21 (75.0)	7 (25.0)		24 (85.7)	3 (10.7)	1 (3.6)	
Increased syncytial knot formations										
Yes	19 (86.4)	3 (13.6)	0.1163	19 (86.4)	3 (13.6)	0.1163	16 (72.8)	3 (13.6)	3 (13.6)	0.6291
No	8 (61.5)	5 (38.5)		8 (61.5)	5 (38.5)		12 (92.3)	1 (7.7)	0	

## Data Availability

Not applicable.

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
