# Peer review of "Placental Cyclophilin A Expression in Pregnancies Complicated with Hypertension"

_ijerph, 2022, doi:10.3390/ijerph19095448_

Round 1

Reviewer 1 Report

This paper, authored by Shaaya et al, compares the expression of Cyclophilin A in in different cell types at feto-maternal interface in normotensive vs hypertensive pregnancies. Although the authors have a reasonable sample size, there are some major concerns.

  • The main question asked in this study is the Cyclophilin A expression. Other than Cyclophilin A expression, the remaining data presented in this study is obvious and has been reported in numerous studies comparing hypertensive vs normotensive pregnancies, and does not add any new information.
  • It is alright to investigate only one question in a study and report that there is no change in cyclophilin expression. However, in such scenario, the methodology applied should be highly convincing which is not the case in this study.
  • The images of immunohistochemistry provided in the manuscript might not be sufficient to support the conclusions drawn in this study. The authors should try co-localization with immunofluorescence to stain placental cells with Cyclophilin A and some cell-specific markers to determine if Cyclophilin is or is not expressed in a specific cell type in placenta.
  • Another possible approach to quantify Cyclophilin A is to isolate different cell types from placenta, extract protein, and run a western blot.

Author Response

Response to reviewers’ comments

Thank you for the constructive and valuable comments.

Reviewer 1

This paper, authored by Shaaya et al, compares the expression of Cyclophilin A in in different cell types at feto-maternal interface in normotensive vs hypertensive pregnancies. Although the authors have a reasonable sample size, there are some major concerns.

  • The main question asked in this study is the Cyclophilin A expression. Other than Cyclophilin A expression, the remaining data presented in this study is obvious and has been reported in numerous studies comparing hypertensive vs normotensive pregnancies, and does not add any new information.

Response: Thank you for the comments. Yes, we agree that the clinical parameters have been well-described in the literature. The aim of this study is to address the expression of Cyclophilin A in placenta comparing between hypertensive and normotensive pregnancies. In the conclusion, although clinical parameter was significantly different, it was not highlighted, as our main focus was on Cyclophilin A, which is lacking in the literature.

  • It is alright to investigate only one question in a study and report that there is no change in cyclophilin expression. However, in such scenario, the methodology applied should be highly convincing which is not the case in this study.

Response: Thank you for the comments. Immunohistochemistry is one of the best methods to determine protein expression as it can localised the site of expression, and therefore can determine which cell actually expresses the protein. This method is widely used by pathologists to determine the cell differentiation/phenotype and tumour differentiation (see references below). Other methods like PCR and Western blot will need to extract RNA and protein from the tissue. However, this will not be able to determine the site of expression. In this study, we intend to evaluate the exact location of Cyclophilin A expression.

David J. Dabbs MD, in Diagnostic Immunohistochemistry. Techniques of Immunohistochemistry: Principles, Pitfalls, and Standardization. 6th edition. 2019.

Idikio HA. Immunohistochemistry in diagnostic surgical pathology: contributions of protein life-cycle, use of evidence-based methods and data normalization on interpretation of immunohistochemical stains. Int J Clin Exp Pathol. 2009;3(2):169-176.

  • The images of immunohistochemistry provided in the manuscript might not be sufficient to support the conclusions drawn in this study. The authors should try co-localization with immunofluorescence to stain placental cells with Cyclophilin A and some cell-specific markers to determine if Cyclophilin is or is not expressed in a specific cell type in placenta.

Response: Thanks for the comments. As placenta pathologists, we are trained to identify the cell types by morphology. We verify the histopathology report of placenta tissue samples by viewing the H&E-stained slides and identification of this cell type is not an issue. In addition, there is no specific immunomarker for individual cell type in placenta.

Example of the textbook used by placental pathologists

1) Placental and Gestational Pathology. (2017). In R. Redline, T. Boyd, & D. Roberts (Eds.), Placental and Gestational Pathology (Diagnostic Pediatric Pathology, p. I). Cambridge: Cambridge University Press.

2) T. Yee Khong, Eoghan E. Mooney, Peter G. J. Nikkels, Terry K. Morgan, Sanne J. Gordijn. Pathology of the Placenta: A Practical Guide 1st ed. 2019 Edition.

  • Another possible approach to quantify Cyclophilin A is to isolate different cell types from placenta, extract protein, and run a western blot.

Response: Yes, we agree, running a western blot will gain more information on the protein expression as a whole for each group. However, to determine the localisation of protein expression will not be possible using western blot. In addition, all our tissues are patients’ samples were fixed in formalin solution. Hence, we were not able to extract protein from the tissue.  

Reviewer 2 Report

The scientific work "Placental Cyclophilin A Expression in Pregnancies complicated with Hypertension" is an interesting work on the expression 14
of cyclophilin A in the placenta of mothers with and without hypertension. The manuscript is well performed, the statistical analysis is well conducted and the language is acceptable. It is not original because a previous work performed by Sun et al. (PMID: 30825933) and cited by the authors is similar, but it was conducted on 82 patients affected by severe preeclampsia and 179 healthy pregnancies; the results of the study of Sun et al. were significant in comparison to this one: could it be related to the study sample and the severity of pathology? in the study of Sun et al., there were patients with severe preeclampsia, in this study only patients with hypertension. The study sample is small and there is also a highly significant difference in the number of cesarean sections: could this aspect influence the results of the study?

Why the authors did not consider the differences between spontaneous pregnancies and IVF pregnancies, as described in previous works? (PMID: 30895253; PMID: 25230734; PMID: 29954233)

The authors have not adequately highlighted the strengths and limitations of their study. I suggest better specifying these points in the discussion and the conclusion of this work.
What are the actual clinical implications of this study? it is important to report the results obtained by the authors in the context of clinical practice and to adequately highlight what contribution this study adds to the literature already existing on the topic (underlining the differences with previous works) and to future study perspectives. 

Author Response

Response to reviewers’ comments

Thank you for the constructive and valuable comments.

The scientific work "Placental Cyclophilin A Expression in Pregnancies complicated with Hypertension" is an interesting work on the expression of cyclophilin A in the placenta of mothers with and without hypertension. The manuscript is well performed, the statistical analysis is well conducted and the language is acceptable.

Response: Thank you for the comments.

It is not original because a previous work performed by Sun et al. (PMID: 30825933) and cited by the authors is similar, but it was conducted on 82 patients affected by severe preeclampsia and 179 healthy pregnancies; the results of the study of Sun et al. were significant in comparison to this one: could it be related to the study sample and the severity of pathology? in the study of Sun et al., there were patients with severe preeclampsia, in this study only patients with hypertension. The study sample is small and there is also a highly significant difference in the number of cesarean sections: could this aspect influence the results of the study?

Response: Thank you for the comments. Although Sun et al. (2019) started with 82 patients with severe preeclampsia, eventually they have tissue samples problem (lost during storage) (see Sun et al. article below) and only 20 severe preeclampsia and 20 healthy placentas were analysed for mRNA and protein expression. Their study sample size was actually less than our study which consists of 35 preeclampsia and 35 normotensive subjects. Furthermore, Sun et al. study did not specify exactly how many samples, the immunofluorescent study was performed. The number of IF study might even be smaller than 20. More importantly, using IF, the cellular details were not clear (see figure in their article). Cyclophilin A staining in which type of cells was not clear. In contrast, our study had all 70 cases stained with cyclophilin A and each cell types were clearly presented. We have added a supplementary table 1, which include all the Cyclophilin A expression pattern in all the 70 cases. This is a finding that have never been reported so far.

See supplementary table 1

Why the authors did not consider the differences between spontaneous pregnancies and IVF pregnancies, as described in previous works? (PMID: 30895253; PMID: 25230734; PMID: 29954233)

Response: Thank you for the comments. Cyclophilin A mainly involved in the pathophysiology of cardiovascular disease, and our aim was to understand its involvement in hypertension in pregnancy. We agree studying its expression in IVF pregnancy may have an interesting outcome. However, this was not our intention. Moreover, our hospital does not have many IVF procedures.

The authors have not adequately highlighted the strengths and limitations of their study. I suggest better specifying these points in the discussion and the conclusion of this work.

Response: Thank you for advice. We have added a paragraph to describe the limitation of this study. See page 9 line 271.

Our study may be limited by the small number of cases. Even thought, the expression CyPA was not significant between the hypertensive and normotensive groups, the p values for fetal endothelial cells and cytotrophoblasts were close to 0.05 (p =0.0869). A large study is warranted to further confirm our findings.

What are the actual clinical implications of this study? it is important to report the results obtained by the authors in the context of clinical practice and to adequately highlight what contribution this study adds to the literature already existing on the topic (underlining the differences with previous works) and to future study perspectives. 

Response: Thank you for pointing out this important aspect of our manuscript. We agree the manuscript lacks information on the clinical implications. We have added a paragraph (see below) describing the clinical implication of determining Cyclophilin A hypertension in pregnancy, in the introduction. See page 2 and line 48 and 75.

Cyclophilin A has been implicated in many cardiovascular diseases such as hypertension, atherosclerosis, atherosclerotic aortic aneurysm, peripheral arterial occlusion, pulmonary arterial hypertension and myocardial hypertrophy. Recent studies have described the potential of drugs specific to Cyclophilin A that may be used in cardiovascular diseases. Therefore, this study could reveal more information on the possible usage of Cyclophilin A as biomarker for predictive of disease and the development of new therapeutic target.

Additional information from review article from Xue et al. (2018). Monoclonal antibody targeting interleukin-1β in coronary artery disease showed reduction in cardiovascular event. Interleukin-1β and Cyclophilin A shared same pro-inflammatory effects on cells in the cardiovascular system. See references.

1) Ridker PM, Everett BM, Thuren T, et al; CANTOS Trial Group. Antiinflammatory therapy with canakinumab for atherosclerotic disease. N Engl J Med. 2017;377:1119–1131.

2) Xue C, Sowden MP, Berk BC. Extracellular and Intracellular Cyclophilin A, Native and Post-Translationally Modified, Show Diverse and Specific Pathological Roles in Diseases. Arterioscler Thromb Vasc Biol. 2018;38(5):986-993.

Round 2

Reviewer 1 Report

The authors have not included any additional data or information to address the concerns raised in the first review.

Author Response

The authors have not included any additional data or information to address the concerns raised in the first review.

Response: Thank you for the comments.

Our study described in details the expression of Cyclophilin A in all the different cell types of placentas. So far, this has not been reported yet; hence, it is a new information.

We agree with additional methods such as PCR (for gene expression) and western blot (for protein expression) will add more information to the study. Regrettably, our study was retrospective and all samples were already fixed by formalin solution. Extraction of RNA and protein from formalin fixed paraffin embedded tissues are technically difficult. We hope it is acceptable that we are unable to perform PCR or western blot.
